# Distinct roles of temporal and frontoparietal cortex in representing actions across vision and language

Moritz F. Wurm[1] & Alfonso Caramazza[1,2]

Both temporal and frontoparietal brain areas are associated with the representation of knowledge about the world, in particular about actions. However, what these brain regions represent and precisely how they differ remains unknown. Here, we reveal distinct functional profiles of lateral temporal and frontoparietal cortex using fMRI-based MVPA. Frontoparietal areas encode representations of observed actions and corresponding written sentences in an overlapping way, but these representations do not generalize across stimulus type. By contrast, only left lateral posterior temporal cortex (LPTC) encodes action representations that generalize across observed action scenes and written descriptions. The representational organization of stimulus-general action information in LPTC can be predicted from models that describe basic agent-patient relations (object- and person-directedness) and the general semantic similarity between actions. Our results suggest that LPTC encodes general, conceptual aspects of actions whereas frontoparietal representations appear to be tied to specific stimulus types.

[1] Center for Mind/Brain Sciences (CIMeC), University of Trento, Corso Bettini 31, 38068 Rovereto, Italy. [2] Department of Psychology, Harvard University, 33 Kirkland Street, Cambridge, MA 02138, USA. Correspondence and requests for materials should be addressed to M.F.W. (email: moritz.wurm@unitn.it)

Our knowledge about things and events in the world is represented at multiple levels, from specific perceptual details (e.g., the movement of a body part) to more general, conceptual aspects (e.g., that a movement serves and is meant to give something to someone). Where these levels are represented in the brain is a central issue in neuroscience but remains unresolved[1–4]. While there is considerable progress in understanding the representation of objects, the representation of action knowledge remains particularly controversial[5–8]. A popular view is that higher-level conceptual aspects of actions are encoded in frontoparietal areas, possibly overlapping with the motor system, whereas perceptual action details such as body parts and movements are encoded in posterior temporal areas in closer proximity to the visual system[8–10]. This conception has recently been challenged by demonstrating that posterior temporal cortex encodes action representations (e.g., of opening and closing) that generalize across a range of perceptual features, such as the body parts[11] and movements used to carry out an action[11,12], the type of object involved in an action[11,12], and whether an action is recognized from photographs or videos[13]. These findings suggest that temporal cortex encodes action representations that abstract away from various details of a perceived action. However, these studies also found that anterior parietal cortex represents actions that generalize across perceptual details (see also ref. [14]). Likewise, both areas are also activated during the semantic processing of action words, which lack specific perceptual details of concrete action exemplars[15,16]. These findings raise a puzzling question: if both posterior temporal and anterior parietal cortex are capable of representing actions at similar, high levels of generality, what are their different roles in recognition and memory? It appears unlikely that the two regions have identical functional profiles and store the same, possibly conceptual-level, information in a duplicate way.

Critically, representations of perceptual details are tied to a specific modality, or stimulus type, whereas conceptual representations are generally accessible via different types of stimuli, e.g., via observation or via reading a text. Neuroimaging studies revealed that understanding actions from observation and from written sentences activates overlapping brain networks in prefrontal and parietal cortex as well as in occipitotemporal brain areas, specifically in posterior middle temporal gyrus (pMTG) and surrounding areas in lateral posterior temporal cortex (LPTC)[17,18,19]. This overlap in activation is usually taken as evidence for the recruitment of the same neural representations accessed during both action observation and sentence comprehension, which would suggest that these representations encode action knowledge at stimulus-independent, conceptual levels. However, overlap in activation is not necessarily due to activation of shared representations[20]. Instead, a brain region may house spatially overlapping but functionally independent neural populations that are each activated via one stimulus type but not the other. To date, it remains unresolved whether any of the identified brain regions represent conceptual aspects of actions that can be accessed by different kinds of stimuli, such as videos or sentences, a necessary condition of conceptual representation.

Here, we applied a more stringent approach, crossmodal multivoxel pattern analyis (MVPA)[21,22], to identify action representations that are action-specific but at the same time generalize across perception and understanding of visual scenes and sentences. By training a classifier to discriminate actions observed in videos and testing the same classifier on its accuracy to discriminate corresponding action sentences, this approach is sensitive to spatially corresponding activation patterns of action videos and sentences, pointing toward action representations that are commonly accessed by both stimulus types. In addition, we used a second, conservative criterion to test whether activation

patterns that generalize across stimulus type are compatible with conceptual representation: neural activity patterns associated with different actions should be less or more similar to each other depending on whether the actions share fewer or more conceptual features with each other. For example, the action of opening should be more similar to closing as compared to taking, and all three actions should be more similar to each other as compared to communicating actions. Hence, if stimulus-independent representations encode conceptual information, then their similarity to each other should not be random but follow semantic principles. We used crossmodal representational similarity analysis (RSA) to test whether semantic models are capable of predicting the similarities of crossmodal action representations, allowing us to specify which aspects of actions are captured by stimulus-general action representations. Following the view that action concepts are represented as propositional structures[23], we hypothesized that stimulus-general action representations encode basic components of action concepts, such as agent–patient relations that describe whether or not an action is directed toward other persons or toward objects[24].

We show that activity patterns in LPTC, but not in frontoparietal cortex, distinguish actions while generalizing across stimulus type. The representational content in LPTC can be predicted by models that capture conceptual aspects of actions. By contrast, action representations in frontoparietal cortex do not generalize across stimulus type. The findings suggest that LPTC encodes conceptual action information that can be accessed independently of stimulus type whereas frontoparietal areas represent action information in a stimulus type-specific manner.

## Results

**Procedure**. In two functional magnetic resonance imaging (fMRI) sessions, participants recognized actions presented in videos and corresponding visually presented sentences (Fig. 1). Participants performed catch trial detection tasks by responding to incomplete or meaningless action videos and grammatically or semantically incorrect sentences (for behavioral results see Supplementary Notes and Supplementary Table 1). The session order was balanced across participants.

**Crossmodal action classification**. Using searchlight analysis[25], we performed two kinds of MVPA to identify brain regions that encode stimulus-general and stimulus-specific action representations. To identify action representations that generalize across stimulus type, we trained a classifier to discriminate actions from video stimuli and tested the classifier on the sentences, and vice versa. This analysis identified a single cluster in the left LPTC, peaking in pMTG, and extending dorsally into posterior superior temporal sulcus (pSTS) and ventrally into inferior temporal gyrus (Fig. 2a, Table 1). By contrast, if classifiers were trained and tested within action videos or within sentences, actions could be discriminated in more extended networks overlapping in occipitotemporal, frontal, and parietal areas (Fig. 2b), in line with previous findings[17,18]. Overall, classification accuracies were higher for videos than for sentences. Apart from these general differences, some areas appeared to be particularly sensitive to observed actions (left and right lateral occipitotemporal cortex and anterior inferior parietal cortex), whereas other areas were particularly sensitive to sentences (left posterior superior temporal gyrus and inferior frontal cortex, anterior temporal lobes). Critically, large parts of frontoparietal cortex discriminated both action videos and sentences, but the absence of crossmodal decoding in these areas suggests that these representations do not generalize across stimulus type.

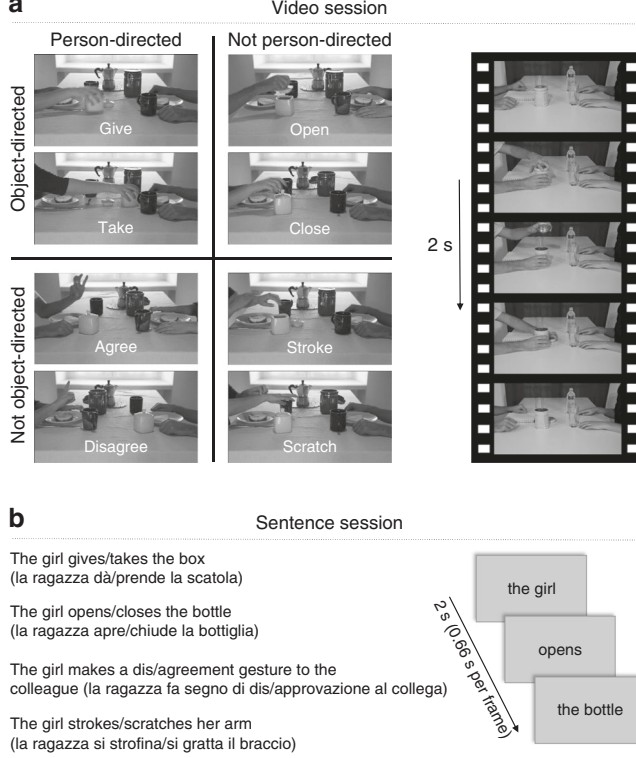

**Fig. 1** Experimental design. The video session (**a**) consisted of action videos (8 actions × 24 video exemplars per action, 2 s per video). The sentence session (**b**) consisted of verbal action descriptions corresponding to the actions shown in the videos (8 actions × 24 sentence exemplars per action, 2 s per sentence)

To investigate the differential effects of crossmodal and within-session decoding in frontoparietal and temporal areas in more detail, we extracted classification accuracies from regions of interest (ROIs) based on the conjunction of the univariate activation maps for action videos vs. baseline and action sentences vs. baseline (Fig. 2c). This conjunction revealed clusters in left inferior frontal gyrus (IFG), left premotor cortex (PMC), left intraparietal sulcus (IPS), and bilateral occipitotemporal cortex extending into LPTC in the left hemisphere. In all ROIs, within-video and within-sentence decoding was significantly above chance, whereas the crossmodal decoding revealed significant effects only in LPTC (Fig. 2d, Supplementary Table 2). To quantify and compare the evidence in the data for $H_1$ (decoding accuracy above chance) and $H_0$ (decoding accuracies not above chance), we performed Bayesian model comparisons using directional Bayesian one-sample $t$-tests[26]. All frontoparietal ROIs revealed moderate evidence for $H_0$ (Bayes factors between 0.11 and 0.21) in the crossmodal decoding (Supplementary Table 2), suggesting that the absence of significant decoding above chance in these areas was unlikely to result from an underpowered design[27] (see also Supplementary Figure 1 for a Bayesian whole-brain analysis). A repeated measures analysis of variance (ANOVA) with the factors ROI and decoding scheme revealed a significant interaction ($F(6, 120) = 9.99$, $p < 0.0001$) as well as main effects for ROI ($F(3, 60) = 18.5$, $p < 0.0001$) and decoding scheme ($F(2, 40) = 59.4$, $p < 0.0001$). Two-tailed paired $t$-tests revealed that crossmodal decoding accuracies in LPTC were significantly higher than in the other ROIs (all $t > 3.5$, all $p < 0.002$). Together, these results further substantiate the view that areas commonly activated during action observation and sentence comprehension may do so on the basis of different principles: while LPTC contains information about action scenes and sentences that can be decoded both within and across stimulus types, frontoparietal areas contain information that can be decoded only within but not across stimulus types.

Could the generalization across stimulus type in left LPTC be explained by verbalization or visual imagery? Our experimental design allowed testing these possibilities: as we balanced the order of video and sentence sessions across participants, we would expect stronger verbalization in the participant group that started with the sentence session, and stronger imagery in the participant group that started with the video session. Contrasting the decoding maps of the two groups, however, revealed no significant differences in left LPTC ($t(19) = 0.04$, $p = 0.97$, two-tailed; Supplementary Figure 2A). In addition, we found no significant correlations between decoding accuracies in LPTC and scores obtained in a post-fMRI rating on verbalization ($r(19) = 0.26$, $p = 0.13$, one-tailed) and visual imagery ($r(19) = -0.31$, $p = 0.97$, one-tailed; Supplementary Figure 2B and C). Together, these control analyses found no support for the hypothesis that visual imagery and verbalization account for the observed crossmodal decoding effects.

Since the actions were directed toward different types of recipients (persons, inanimate objects, and arms/hands), it is possible that classification was driven by information about the recipients alone rather than information specific for actions. To test whether the identified cluster in LPTC overlapped with brain areas associated with the visual processing of different object categories similar to the action recipients of our study, we compared the extent of the LPTC cluster with peak locations of localizers for whole bodies, hands, and tools[28–30]. Areas responding to these object categories were located more ventrally and posteriorly of LPTC and did not substantially overlap with LPTC (Supplementary Figure 3). We also investigated the overlap with areas capable of discriminating object categories across modality as identified using crossmodal MVPA[21,31]. These effects were located more ventrally and anteriorly in inferior temporal cortex and also showed no substantial overlap with LPTC (Supplementary Figure 3). Taken together, these findings do not support the assumption that crossmodal action classification was driven by information specifically about objects rather than actions.

**Crossmodal multiple regression RSA.** The accessibility of action-specific representations during both action observation and sentence reading suggests a central role of LPTC in action understanding. What exactly do these representations encode? Using multiple regression RSA, we analyzed the structure of action representations in LPTC in more detail. To this end, we extracted, for each participant, the pairwise crossmodal classification accuracies to construct neural dissimilarity matrices, which reflect how well the actions could be discriminated from each other, and thus how dissimilar the action representations are to each other (Fig. 3a). We found that the neural action dissimilarity in LPTC could be predicted by models of person- and object-directedness, which were based on post-fMRI ratings of how much the actions shown in the experiment took into account the reactions of other persons (person-directedness) and how much the actions involved an interaction with physical, nonliving objects (object-directedness, Fig. 3b). Additional variance in the neural representational similarity could be explained by a more general model of semantic action similarity that group actions based on a combination of semantic action relations[32]. Other models that were included in the regression to control for factors of no interest (semantic object similarity, task difficulty, unusualness of actions) could not explain further variance.

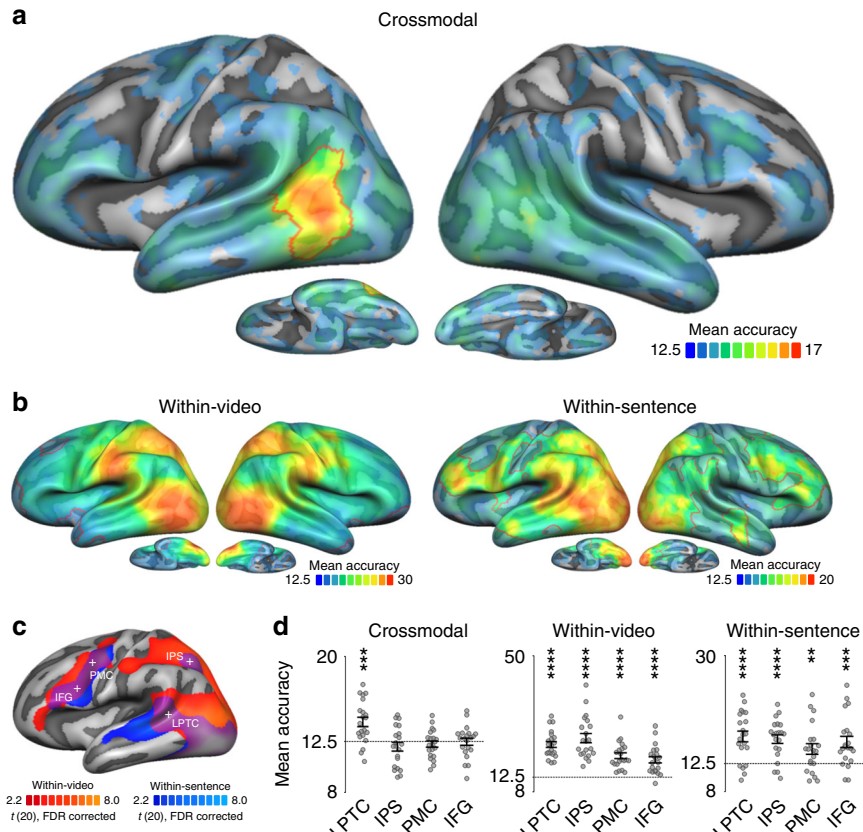

**Fig. 2** Multiclass action decoding searchlight (mean accuracies, chance = 12.50%). Mean accuracy maps of crossmodal (**a**) and within-modality (**b**) classifications. Red outlines indicate TFCE-corrected areas; uncorrected areas are displayed to indicate putative trends for above chance decoding in areas not surviving correction (for clarity, higher transparency is applied to uncorrected areas). **c** Overlap of univariate activation maps for the contrasts action videos vs. baseline and action sentences vs. baseline (FDR-corrected). Spherical ROIs (12 mm radius) were centered on conjunction peaks (indicated by crosses). **d** ROI decoding accuracies for crossmodal and within-modality classification. Asterisks indicate FDR-corrected effects: **p < 0.01, ***p < 0.001, ****p < 0.0001. Error bars/lines indicate SEM. IFG inferior frontal gyrus, IPS intraparietal sulcus, LPTC lateral posterior temporal cortex, PMC premotor cortex

RSA effects were robust across ROI size and number of models included in the analysis (Supplementary Figure 4) and were not modulated by effects of session order, verbalization, or visual imagery (Supplementary Figure 5). In addition, we found that the rating-based models of person- and object-directedness correlated significantly better with the neural action dissimilarities than simple categorical models that were based on the main stimulus dimensions used in the experiment (Fig. 3c; person-directedness: $Z = 3.62$, $p = 0.0003$, object-relatedness: $Z = 3.01$, $p = 0.002$; two-tailed signed-rank test). This suggests that the neural representational organization in LPTC indeed reflect stimulus variations in person- and object-directedness as measured by behavioral judgments rather than other hidden factors that may have accidentally covaried with the stimulus dimensions. Finally, we tested whether models based on individual ratings outperform models based on group-averaged ratings. Correlations with neural dissimilarities were slightly higher for group-averaged as compared to individual models, but differences were not significant (person-directedness: $Z = 1.59$, $p = 0.11$, object-relatedness: $Z = 0.29$, $p = 077$; two-tailed signed-rank test).

**Path-of-ROI RSA.** Crossmodal action information could be decoded from a relatively large cluster in LPTC spanning from pSTS to inferior temporal gyrus. Is this area parceled into distinct subregions specialized for certain aspects of actions? Based on previous findings[24], we hypothesized that conceptual information

related to person- and object-directedness is not distributed uniformly in LPTC but rather follows a distinctive functional topography that parallels known gradients in more posterior areas segregating animate and inanimate object knowledge[33,34] and biological and functional tool motion[35]. To investigate how the representational content changes from dorsal to ventral temporal cortex, we plotted the performance of the person-directedness and object-relatedness models as a function of position along a cortical vector from the dorsal to the ventral end of the crossmodal decoding cluster (Fig. 3d). In line with our prediction, we found that the person-directedness model explained most variance in dorsal LOTC at the level of pSTS (Talairach $z = 9$), whereas the object-directedness model peaked more ventrally (Talairach $z = 3$) at the level of pMTG (ANOVA interaction position × model: $F(15, 300) = 4.23$, $p < 0.001$).

**Discussion**

Conceptual representations should generally be accessible independently of the modality or type of stimulus, e.g., whether one observes an action or reads a corresponding verbal description. Here we show that overlap of activation alone during action observation and sentence comprehension cannot be taken as evidence for the access of stimulus-independent, conceptual representations. Using crossmodal MVPA, we found that only the left LPTC reveals neural activity patterns that are both specific for distinct actions and at the same time generalize across action

**Table 1 Clusters identified in the crossmodal, within-sentence, and within-video action decoding**

| Region | x | y | z | t | p | Accuracy |
|---|---|---|---|---|---|---|
| *Crossmodal* | | | | | | |
| Left pMTG | −54 | −61 | 4 | 7.78 | 8.94E−08 | 17.7 |
| Left pSTS | −48 | −49 | 10 | 6.91 | 1.79E−07 | 16.9 |
| *Within-video* | | | | | | |
| Left LOTC | −45 | −64 | 1 | 18.24 | 3.11E−14 | 35.5 |
| Right LOTC | 45 | −58 | 4 | 15.92 | 3.99E−13 | 35.0 |
| Left aIPL | −54 | −28 | 31 | 14.61 | 1.95E−12 | 31.0 |
| Right aIPL | 54 | −22 | 31 | 13.11 | 1.40E−11 | 27.5 |
| Right IPS | 24 | −55 | 43 | 12.70 | 2.50E−11 | 31.1 |
| Left IPS | −27 | −79 | 22 | 11.00 | 3.09E−10 | 30.4 |
| *Within-sentence* | | | | | | |
| Left LOTC | −45 | −55 | 1 | 7.03 | 1.00E−06 | 21.1 |
| Right LOTC | 45 | −55 | −2 | 7.59 | 1.31E−07 | 18.3 |
| Left IPS | −27 | −58 | 43 | 7.00 | 1.00E−06 | 19.7 |
| Right IPS | 12 | −70 | 34 | 8.09 | 4.93E−08 | 19.1 |
| Left LOC | −21 | −85 | 1 | 11.30 | 1.96E−10 | 26.9 |
| Right LOC | 24 | −82 | 1 | 9.36 | 4.75E−09 | 23.9 |
| Left pSTG | −45 | −40 | 31 | 5.50 | 2.20E−05 | 20.5 |
| Left VWFA | −42 | −52 | −20 | 6.74 | 1.00E−06 | 19.7 |
| Left IFG/IFS | −42 | 23 | 22 | 5.48 | 2.30E−05 | 19.5 |
| Right IFG/MFG | −42 | 23 | 22 | 5.48 | 2.30E−05 | 19.5 |
| Left PMC | −54 | 2 | 31 | 5.34 | 3.10E−05 | 19.3 |
| Right IFJ | 33 | 8 | 34 | 6.96 | 1.00E−06 | 18.0 |
| Left ATL | −60 | −13 | −5 | 4.81 | 1.06E−04 | 18.3 |
| Right ATL | 60 | −13 | −2 | 4.00 | 7.12E−04 | 17.8 |

Coordinates (x, y, z) in Talairach space. Decoding accuracy at chance is 12.5%. Maps were TFCE corrected. For the within-video and within-sentence action decoding, only main clusters with distinctive peaks are reported
aIPL anterior inferior parietal lobe, ATL anterior temporal lobe, IFG inferior frontal gyrus, IFJ inferior frontal junction, IFS inferior frontal sulcus, IPS intraparietal sulcus, LOC lateral occipital cortex, LOTC lateral occipitotemporal cortex, MFG middle frontal gyrus, PMC premotor cortex, pMTG posterior middle temporal gyrus, pSTG posterior superior temporal gyrus, pSTS posterior superior temporal sulcus, VWFA visual word form area

scenes and sentences. What accounts for the action-specific correspondence between action scenes and sentences in LPTC?

One possibility is that crossmodal decoding was due to visual imagery (during reading the sentences) or verbalization (during watching the action scenes). If so, the decoded information in LPTC was in a verbal format, triggered by both the sentences and by verbalizations of the action scenes, or in a visual format, triggered by the observed action scenes and imagery of the sentences. However, control analyses do not support this possibility: the strength of crossmodal decoding was not influenced by session order, by the participants' tendencies to verbalize or to imagine the actions, or by the strength of correspondence between verbalizations and sentences or between imagined and observed actions. Nonetheless, it is possible that verbalization or imagery were so implicit and automatic that they were not captured by the participants' subjective ratings, that session order did not substantially influence the tendencies to verbalize or imagine the actions, or that retrospection of previously experienced action scenes and sentences had no measurable impact on verbalization or imagery. However, these assumptions would not explain why effects of verbalization or imagery resulted in a match between action videos and sentences in left LPTC only and not in other brain regions that were found in the within-video or the within-sentence decoding. A second possibility is that neural populations carrying action-specific information for the two stimulus types are functionally independent but lie next to each other within individual voxels. While this scenario is possible, it would raise the question about the purpose of such voxel-by-voxel correspondence between representations of action scenes and sentences and why left LPTC is the only area with this representational profile. Together, these objections do not dispute that left LPTC reveals a representational profile that is fundamentally different than those found in frontoparietal areas. In

consideration of the neuroimaging[16], neuropsychological[36], and virtual lesion[37] evidence we have about this area (see ref. [38] for a review), the most plausible interpretation seems to be that left LPTC encodes a conceptual level of representation that can be accessed by different modalities and stimulus types like action scenes and sentences. This interpretation is further supported by crossmodal RSA, which revealed that the organization of stimulus-general information in LPTC could be predicted by models that describe basic conceptual aspects of actions, i.e., agent–patient relations (person- and object-directedness) as well as more complex semantic relations between action concepts such as whether actions are subparts of or in opposition with each other. Note that the conclusion reached here concerns details of the information represented in LPTC and is silent on ongoing debates about the precise representational format[20,39].

Frontoparietal areas discriminated both action scenes and sentences, but the decoded representations did not generalize across the two stimulus types. Representations in LPTC thus seem to be more general and abstract as compared to frontoparietal representations, which appear to capture more specific details and properties of the different stimulus types. Notably, observed actions are more specific and rich in details compared to sentences. The stronger decoding of action scenes relative to sentences appears to reflect this difference in richness of detail. The decoded frontoparietal representations might capture information about how specifically an action is carried out. Such motor-related representations might be triggered by action scenes reflecting specific aspects of the action[36,40], such as the kinematics of an action or the particular grip used on an object, whereas the motor-related representations triggered by sentences would be less specific, more variable, and less robust. Following this view, frontoparietal motor-related representations are activated following conceptual activation in LPTC, in line with the finding

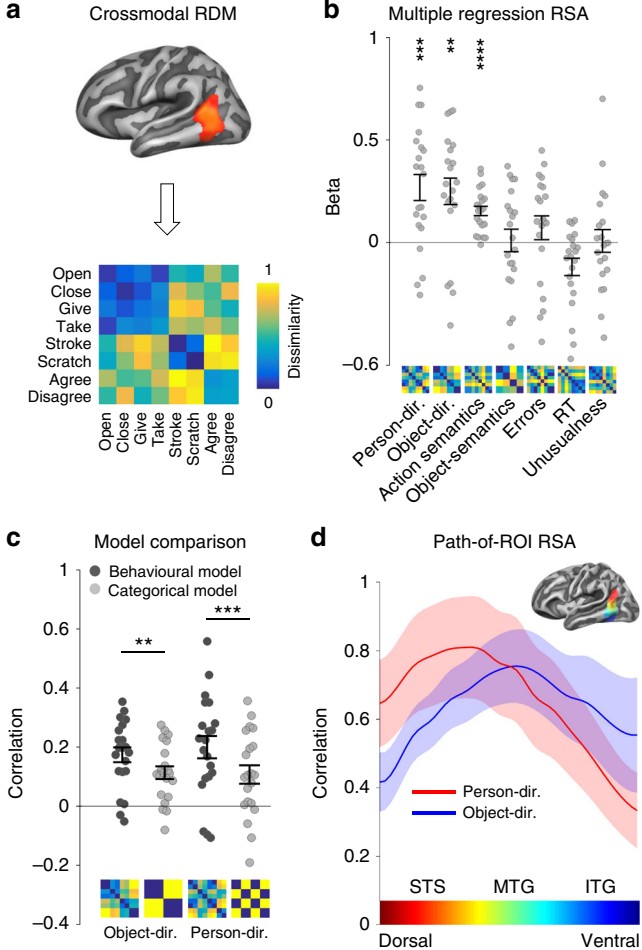

**Fig. 3** Crossmodal RSA. **a** Classification matrices were extracted from left LPTC, averaged, and converted into a dissimilarity matrix for each participant (see Methods for details; displayed matrix averaged across participants, rank-transformed, and scaled[66]). **b** Multiple regression RSA. Person- and object-directedness were based on ratings; WordNet action and object models were based on taxonomical distance (path length); crossmodal control models were based on behavioral experiments (errors, reaction times), and ratings for unusualness of actions presented in videos and sentences. **c** Comparison between performance of behavioral (rating-based) and categorical models of person- and object-directedness using correlation-based RSA. **d** Path-of-ROI analysis. Mean RSA correlation coefficients plotted as a function of position along dorsal–ventral axis. x axis color bar corresponds to ROI colors. Asterisks in **b** and **c** indicate FDR-corrected effects: **$p < 0.01$, ***$p < 0.001$, ****$p < 0.0001$. Error bars/lines indicate SEM

that perturbation of pMTG induced by transcranial magnetic stimulation not only impedes semantic processing of action verbs but also disrupts increased motor excitability for motor as compared to non-motor verbs[37]. Likewise, dysplasic individuals born without arms recognize hand actions, for which they do not have corresponding motor representations, as fast and as accurately as typically developed participants suggesting that motor-related representations are not necessary in accessing action concepts[41]. Notably, neural activity and representational content in fronto-parietal cortex does not differ between dysplasic and typically developed individuals during hand action observation[11]. An additional possibility is therefore that frontoparietal areas are involved in processing non-motor information related to the stimuli, for example in the service of anticipating how action

scenes and sentences unfold[42–44]. In support of this view, pre-motor and parietal areas have been shown to be engaged in the prediction of dynamic stimuli of different types and modalities[45], even if stimuli lack any motor-relevant pragmatic properties[46].

Whereas the crossmodal decoding analyses reported here focused on identifying the neural substrate of stimulus-general action representations, the crossmodal RSA allowed us to investigate the structure of these representations in more detail. We found that the similarity of neural patterns associated with the actions tested in this experiment could be predicted by models that describe whether actions are directed toward other persons or toward inanimate objects as well as a more general model of semantic action similarity. Since person- and object-directedness alone do not fully capture the meaning of an action, it is not surprising that LPTC appears to encode additional information about semantic aspects of actions. It remains to be determined what precisely these additional aspects are. Our study was designed to investigate representations that capture the action directedness toward other persons and inanimate objects. Future studies should test other relevant dimensions (e.g., directedness to the acting person's own body, change of location in space) and test for additional organizational principles, which were only partly captured by the semantic action similarity model, for example, aspects of causative order (that determine the difference between open and close) or thematic relations (that capture whether actions are part of a common overarching activity as in aim and shoot).

Notably, representations sensitive to person- and object-directedness followed distinctive functional topographies: neural populations in the dorsal part of LPTC were more sensitive to detect whether an action is person-directed or not, whereas neural populations in more ventral parts were more sensitive to detect whether an action is object-directed or not. This result replicates the previous observation of a dorsal–ventral gradient for observed actions[24] and demonstrates that a similar (but left-lateralized) gradient exists also for stimulus-general action representations. The topographical distinction of person- and object-directedness resembles related gradients that are typically found in adjacent posterior/ventral areas: in lateral occipito-temporal cortex, overlapping with the posterior part of the cluster found in the present study, dorsal subregions are preferentially activated by animate objects (e.g., animals and body parts), whereas ventral subregions are preferentially activated by inani-mate objects (e.g., manipulable artifacts and tools)[33,34]. Likewise, dorsal subregions are preferentially activated by body movements, whereas ventral subregions are preferentially activated by action-specific tool movements[35]. Here we demonstrate a continuation of this distinction for more complex action components that cannot be explained by visual stimulus properties. The topo-graphic alignment of object, object motion, and modality-general action representation points to an overarching organizational principle of action and object knowledge in temporal and occi-pital cortex: Knowledge related to persons is represented in dorsal posterior occipitotemporal cortex and specifically associated with (and form the perceptual basis of) biological motion and person-directed actions in dorsal LPTC. Furthermore, knowledge about inanimate manipulable objects is represented in ventral occipi-totemporal cortex and specifically associated with action-related object motion patterns and object-directed actions in ventral LPTC. The functionally parallel organization of object and action knowledge in lateral occipital and temporal cortex, respectively, suggest the important role of domain-specific object-action con-nections along the posterior-anterior axis, in agreement with the view that connectivity plays a fundamental role in shaping the functional organization of distinct knowledge categories[47,48]. Following this view, LPTC encodes representations that integrate

more basic precursors (such as information about persons and objects) from adjacent posterior areas. This view is supported by the finding that LPTC is activated for object pairs that are thematically related, i.e., are linked by an action or event (e.g. saw-wood, squirrel-nut) as compared to object pairs that are taxonomically related, i.e., belong to the same object category (e.g. saw-hammer, squirrel-dog), which activate more posterior areas in occipital cortex[49] (but note that taxonomic object relations are also associated with anterior temporal lobe; see ref. [50] for a review).

In conclusion, our results demonstrate that overlap in activation does not necessarily indicate recruitment of a common representation or function and that cross-decoding is a powerful tool to detect the presence or absence of representational correspondence in overlapping activity patterns. Specifically, we revealed fundamental differences between frontoparietal and temporal cortex in representing action information in stimulus-dependent and -independent manners. This result may shed new light onto the representational profiles of frontoparietal and temporal regions and their roles in semantic memory. The different levels of generality in these areas point to a hierarchy of action representation from specific perceptual stimulus features in occipitotemporal areas to more general, conceptual aspects in left LPTC, and back to stimulus-specific representations in frontoparietal cortex. We propose that the topographic organization of conceptual action knowledge is (at least partially) determined by the representation of more basic precursors, such as animate and inanimate entities.

## Methods

**Participants**. Twenty-two right-handed native Italian speakers (nine females; mean age, 23.8 years; age range, 20–36 years) participated in this experiment. All participants had normal or corrected-to-normal vision and no history of neurological or psychiatric disease. One subject was excluded due to poor behavioral performance in the task (accuracy 2 standard deviations below the group mean). All procedures were approved by the Ethics Committee for research involving human participants at the University of Trento, Italy.

**Stimuli**. The video stimuli consisted of 24 exemplars of eight hand actions (192 action videos in total) as used in Wurm et al.[24]. The actions varied along two dimensions, person-directedness (here defined as the degree to which actions take into account the actions and reactions of others) and object-directedness (here defined as the degree to which actions involve the interaction with physical inanimate objects), resulting in four action categories: change of possession (object-directed/person-directed): give, take; object manipulation (object-directed/non-social): open, close; communication (not object-directed/person-directed): agree, disagree; body/contact action (not object-directed/not person-directed): stroke, scratch. By using 24 different exemplars for each action, we increased the perceptual variance of the stimuli to ensure that classification is trained on conceptual rather than perceptual features. Variance was induced by using two different contexts, three perspectives, two actors, and six different objects that were present or involved in the actions (kitchen context: sugar cup, honey jar, coffee jar; office context: bottle, pen box, aluminum box). Video catch trials consisted of six deviant exemplars of each of the eight actions (e.g., meaningless gestures or object manipulations, incomplete actions; 48 catch trial videos in total). All 240 videos were identical in terms of action timing, i.e., the videos started with hands on the table, followed by the action, and ended with hands moving to the same position on the table. Videos were gray scale, had a length of 2 s (30 frames per second), and a resolution of 400 × 225 pixels.

The sentence stimuli were matched with the video stimuli in terms of stimulus variance (24 sentences of eight actions; 192 sentence videos in total). All sentences had the structure subject-verb-object. For each action, we first defined the corresponding verb phrase: dà (s/he gives), prende (s/he takes), apre (s/he opens), chiude (s/he closes), si strofina (s/he rubs her/his), si gratta (s/he scratches her/his), fa segno di approvazione a (s/he makes a sign of agreement to), fa segno di disapprovazione a (s/he makes a sign of disagreement to). To create 24 exemplars per action, we crossed each verb phrase with six subjects: lei, lui, la ragazza, il ragazzo, la donna, l'uomo (she, he, the girl, the boy, the woman, the man) and with four objects, which matched the objects used in the videos as much as possible. Change of possession: la scatola, il vaso, il caffe, lo zucchero (the box, the jar, the coffee, the sugar); object manipulation: la bottiglia, il barattolo, la cassetta, l'astuccio (the bottle, the can, the casket, the pencil case); communication: l'amica, l'amico, il college, la collega (friend, colleague); body action: il braccio, la mano, il

gomito, l'avambraccio (the arm, the hand, the elbow, the forearm). As the crossmodal analysis focuses on the generalization across sentence and video stimuli, perceptual and syntactic differences between action sentences, such as sentence length and occurrence of prepositions, were ignored. Catch trial sentences consisted of six grammatically incorrect or semantically odd exemplars of each of the eight actions (e.g., lei apre alla bottiglia (she opens to the bottle), lui dà l'amica (he gives the friend); 48 catch trial sentences in total). The sentences were presented superimposed on light gray background (400 × 225 pixels) in three consecutive chunks (subject, verb phrase, object), with each chunk shown for 666 ms (2 s per sentence), using different font types (Arial, Times New Roman, Comic Sans MS, MV Boli, MS UI Gothic, Calibri Light) and font sizes (17–22) to increase the perceptual variance of the sentence stimuli (balanced across conditions within experimental runs).

In the scanner, stimuli were back-projected onto a screen (60 Hz frame rate, 1024 × 768 pixels screen resolution) via a liquid crystal projector (OC EMP 7900, Epson Nagano, Japan) and viewed through a mirror mounted on the head coil (video presentation 6.9° × 3.9° visual angle). Stimulus presentation, response collection, and synchronization with the scanner were controlled with ASF[51] and the Matlab Psychtoolbox-3 for Windows[52].

**Experimental design**. For both video and sentence sessions, stimuli were presented in a mixed event-related design. In each trial, videos/sentences (2 s) were followed by a 1 s fixation period. Eighteen trials were shown per block. Each of the nine conditions (eight action conditions plus one catch trial condition) was presented twice per block. Six blocks were presented per run, separated by 10 s fixation periods. Each run started with a 10 s fixation period and ended with a 16 s fixation period. In each run, the order of conditions was first-order counter-balanced[53]. Each participant was scanned in two sessions (video and sentence session), each consisting of four functional scans, and one anatomical scan. The order of sessions was counterbalanced across participants (odd IDs: videos-sentences, even IDs: sentences-videos). For each of the nine conditions per session, there was a total of 2 (trials per block) × 6 (blocks per run) × 4 (runs per session) = 48 trials per condition. Each of the 24 exemplars per action condition was presented twice in the experiment.

**Task**. Before fMRI, we instructed and trained participants for the first session only (videos or sentences). The second session was instructed and practiced within the scanner after the four runs of the first session. Participants were asked to attentively watch the videos (read the sentences) and to press a button with the right index finger on a response button box whenever an observed action was meaningless or performed incompletely or incorrectly (whenever a sentence was meaningless or grammatically incorrect). The task thus induced the participants to understand the actions, while minimizing the possibility of additional cognitive processes that might be different between the actions but similar across sessions. For example, tasks that require judgments about the actions along certain dimensions such as action familiarity[54] might lead to differential neural activity related to the preparation of different responses, which could be decodable across stimulus type. Participants could respond either during the video/sentence or during the fixation phase after the video/sentence. To ensure that participants followed the instructions correctly, they completed a practice run before the respective session. Participants were not informed about the purpose and design of the study before the experiment.

**Post fMRI survey**. After the fMRI session, participants judged the degree of person- and object-directedness of the actions in the experiment. For each action, participants answered ratings to the following questions: object-directedness: "How much does the action involve an interaction with a physical, inanimate object?" Person-directedness: "How much does the action take into account the actions and reactions of another person?" In addition, they were asked to estimate how much they verbalized the actions during the video session (verbalization: "During watching the action videos, did you verbalize the actions, that is, did you have verbal descriptions (words, sentences) in your mind as if you were talking to yourself?"), how much they visually imagined the action in the sentence session (imagery: "During reading the sentences, did you visually imagine concrete action scenes?"), and how similar their verbal descriptions were to the sentences (verbalization-sentence correspondence) and how similar the imagined action scenes were to the videos; (imagery-video correspondence, respectively). For all ratings, 6-point Likert scales (from 1 = not at all to 6 = very much) were used.

**Representational dissimilarity models**. To investigate the representational organization of voxel patterns that encode crossmodal action information, we tested the following models of representational dissimilarity:

To generate models of person- and object-directedness, we computed pairwise Euclidean distances between the group-averaged responses to each of the actions from the ratings for person- and object-directedness. For comparison, we also tested categorical models that segregated the actions along person- and object-directedness without taking into account more subtle action-specific variations.

To test an exploratory model of semantic relationship between the actions (action semantics hereafter) that is not solely based on either person- or

object-directedness, but rather reflect semantic relations between action concepts, we computed hierarchical distances between action concepts based on WordNet 2.1[55]. This model captures a combination of semantic relations between actions, such as whether actions are subparts of each other (e.g., drinking entails swallowing) or oppose each other (e.g., opening and closing). We used WordNet because it is supposed to reflect conceptual-semantic rather than syntactic relations between words. Semantic relations should therefore be applicable to both action sentences and videos. We selected action verbs by identifying the cognitive synonyms (synsets) in Italian that matched the verbs used in the action sentences and that corresponded best to the meaning of the actions ("open.v.01," "close.v.01," "give.v.03," "take.v.08," "stroke.v.01," "scratch.v.03," "agree.v.01," "disagree.v.01"). We computed pairwise semantic distances between the eight actions using the shortest path length measure, which reflects the taxonomical distance between action concepts.

In a similar way, we generated a model of semantic relationship between the target objects (inanimate objects, body parts, persons) of the actions in the four action categories (object semantics). As for the action verbs, we selected Italian synsets that matched the object nouns in the sentences. As there were four objects per action category, the distances were averaged within each category.

To generate models of task difficulty (RT and errors), an independent group of participants ($N = 12$) performed a behavioral two-alternative forced choice experiment that had the same design and instruction as the fMRI experiment except that participants responded with the right index finger to correct action videos/sentences (action trials) and with the right middle finger to incorrect action videos/sentences (catch trials). Errors and RTs of correct responses to action trials were averaged across participants. The error model was constructed by computing the pairwise Euclidean distances between the eight accuracies of the sentence session and the eight accuracies of the video session. The model thus reflects how similar the eight actions are in terms of errors made across the two sessions. The RT model was constructed in a similar way except that the RTs of each session were z-scored before computing the distances to eliminate session-related differences between video and sentence RTs.

To generate a model reflecting potential differences in saliency due to unusualness between the actions, we asked the participants of the behavioral experiment described in the previous paragraph to judge how unusual the actions were in the videos and sentences, respectively (using 6-point Likert scales from 1 = not at all to 6 = very much). The unusualness model was constructed by computing the pairwise Euclidean distances between the eight mean responses to the sentence session and the eight mean responses to the video session.

**Data acquisition**. Functional and structural data were collected using a 4 T Bruker MedSpec Biospin MR scanner and an eight-channel birdcage head coil. Functional images were acquired with a T2*-weighted gradient echo-planar imaging (EPI) sequence with fat suppression. Acquisition parameters were a repetition time (TR) of 2.2 s, an echo time of 33 ms, a flip angle (FA) of 75°, a field of view (FOV) of 192 mm, a matrix size of 64 × 64, and a voxel resolution of 3 × 3 × 3 mm. We used 31 slices, acquired in ascending interleaved order, with a thickness of 3 mm and 15% gap (0.45 mm). Slices were tilted to run parallel to the superior temporal sulcus. In each functional run, 176 images were acquired. Before each run, we performed an additional scan to measure the point-spread function (PSF) of the acquired sequence to correct the distortion expected with high-field imaging[56].

Structural T1-weigthed images were acquired with an MPRAGE sequence (176 sagittal slices, TR = 2.7 s, inversion time = 1020 ms, FA = 7°, 256 × 224 mm FOV, 1 × 1 × 1 mm resolution).

**Preprocessing**. Data were analyzed using BrainVoyager QX 2.8 (BrainInnovation) in combination with the BVQXTools and NeuroElf Toolboxes and custom software written in Matlab (MathWorks). Distortions in geometry and intensity in the echo-planar images were corrected on the basis of the PSF data acquired before each EPI scan[57]. The first four volumes were removed to avoid T1 saturation. The first volume of the first run was aligned to the high-resolution anatomy (six parameters). Data were three-dimensional (3D) motion corrected (trilinear interpolation, with the first volume of the first run of each participant as reference), followed by slice time correction and high-pass filtering (cutoff frequency of three cycles per run). Spatial smoothing was applied with a Gaussian kernel of 8 mm full with at half maximum (FWHM) for univariate analysis and 3 mm FWHM for MVPA. Anatomical and functional data were transformed into Talairach space using trilinear interpolation.

**Action classification**. For each participant, session, and run, a general linear model was computed using design matrices containing 16 action predictors (2 predictors per action, each based on 6 trials selected from the first half (blocks 1–3) or the second half (blocks 4–6) of each run), catch trials, and of the 6 parameters resulting from 3D motion correction (x, y, z translation and rotation). Each predictor was convolved with a dual-gamma hemodynamic impulse response function[58]. Each trial was modeled as an epoch lasting from video/sentence onset to offset (2 s). The resulting reference time courses were used to fit the signal time courses of each voxel. In total, this procedure resulted in 8 beta maps per action condition and session.

Searchlight classification[25] was performed for each participant separately in volume space using searchlight spheres with a radius of 12 mm and a linear support vector machine (SVM) classifier as implemented by the CoSMoMVPA toolbox[59] and LIBSVM[60]. We demeaned the data for each multivoxel beta pattern in a searchlight sphere across voxels by subtracting the mean beta of the sphere from each beta of the individual voxels. Demeaning was done to minimize the possibility that classifiers learn to distinguish actions based on global univariate differences in a ROI that could arise from different processing demands within each stimulus type due to non-conceptual differences between actions (e.g., some action scenes might contain more vs. less motion information, differences in sentence length). In all classification analyses, each action was discriminated from each of the remaining seven actions in a pairwise manner ("one-against-one" multiclass decoding). For searchlight analyses, accuracies were averaged across the eight actions (accuracy at chance = 12.5%) and assigned to the central voxel of each searchlight sphere. In the crossmodal action classification, we trained a classifier to discriminate the actions using the data of the video session and tested the classifier on its accuracy at discriminating the actions using the data of the sentence session. The same was done vice versa (training on sentence data, testing on video data), and the resulting accuracies were averaged across classification directions. We also tested whether the generalization order matters, i.e., whether generalization from action videos (training) to sentences (testing) resulted in different clusters than generalization from action sentences (training) to videos (testing). However, both generalization schemes resulted in similar maps and contrasting the generalization schemes using paired t-tests revealed no significant clusters. In the within-video action classification, we decoded the eight actions of the video session using leave-one-beta-out cross validation: we trained a classifier to discriminate the actions using seven out of the eight beta patterns per action. Then we tested the classifier on its accuracy at discriminating the actions using the held out data. This procedure was carried out in eight iterations, using all possible combinations of training and test patterns. The resulting classification accuracies were averaged across the eight iterations. The same procedure was used in the within-sentence action classification. Individual accuracy maps were entered into a one-sample t-test to identify voxels in which classification was significantly above chance. Statistical maps were thresholded using threshold-free cluster enhancement (TFCE;[61] as implemented in the CoSMoMVPA Toolbox[59]). We used 10000 Monte Carlo simulations and a one-tailed corrected cluster threshold of $p = 0.05$ ($z = 1.65$). Maps were projected on a cortex-based aligned group surface for visualization. Significant decoding accuracies below chance (using both two- and one-tailed tests) were not observed.

**ROI analysis**. To specifically investigate differential effects of crossmodal and within-session decoding in frontoparietal and posterior temporal areas that are commonly activated during action observation and sentence comprehension, we performed a ROI analysis based on the conjunction[62] of the FDR-corrected RFX contrasts action videos vs. baseline and action sentences vs. baseline. ROIs were created based on spheres with 12 mm radius around the conjunction peaks within each area (Talairach coordinates x/y/z; IFG: −42/8/22, PMC: −39/−7/37, IPS: −24/−58/40, LPTC: −45/−43/10; no conjunction effects were observed in the right hemisphere except in ventral occipitotemporal cortex). From each ROI, classification scheme (crossmodal, within-video, within-sentence), and participant, decoding accuracies were extracted from the searchlight maps and averaged across voxels. Mean decoding accuracies were entered into ANOVA, one-tailed one-sample t-tests, and Bayesian comparisons[26]. Bayes factors were computed using R (version 3.4.2) and the BayesFactor package (version 0.9.12)[63]. Using one-sided Bayesian one-sample t-tests, we computed directional Bayes factors to compare the hypotheses that the standardized effect is at chance (12.5%) vs. above chance, using a default Cauchy prior width of $r = 0.707$ for effect size. Bayes factor maps were computed using the same procedure for each voxel of the decoding maps.

**Crossmodal representational similarity analysis**. For further investigation of the representational organization in the voxels that encode crossmodal action information, we performed an ROI-based multiple regression RSA[28,64] using crossmodal classification accuracies[21]. ROIs were defined based on TFCE-corrected clusters identified in the classification analysis. From each ROI cluster, we extracted the pairwise classifications of the crossmodal action decoding, resulting in voxel-wise 8 × 8 classification matrices. Notably the selection of action-discriminative voxels, which is based on on-diagonal data of the pairwise classification matrix, does not bias toward certain representational organizations investigated in the RSA, which is based on off-diagonal data of the pairwise classification matrix (see below). Classification matrices were averaged across voxels, symmetrized across the main diagonal, and converted into representational dissimilarity matrices (RDM) by subtracting 100—accuracy (%), resulting in one RDM per ROI and participant.

The off-diagonals (i.e., the lower triangular parts) of the individual neural and the model RDMs were vectorized, z-scored, and entered as independent and dependent variables, respectively, into a multiple regression. We tested for putative collinearity between the models by computing condition indices (CI), variance inflation factors (VIFs), and variance decomposition proportions (VDP) using the colldiag function for Matlab. The results of these tests (max. CI = 3, max. VIF = 2.6, max VDP = 0.8) revealed no indications of potential estimation problems[65]. Correlation coefficients were Fisher transformed and entered into one-

tailed signed-rank tests. Results were FDR-corrected for the number of models included in the regression.

To compare the performance of individual models, we used correlation-based RSA, i.e., we computed rank correlations between the off-diagonals of neural and model RDMs using Kendall's $\tau_A$ as implemented by the toolbox for RSA[66]. Comparisons between rank correlations were computed using FDR-corrected paired two-tailed signed-rank tests.

To investigate the performance of models along the dorsal–ventral axis, we used a path-of-ROI analysis[24,34]: Dorsal and ventral anchor points were based on the most dorsal and ventral voxels, respectively, of the cluster identified in the crossmodal action classification (Talairach $x/y/z$; dorsal: $-43/-58/20$; ventral: $-43/-50/-10$). Anchor points were connected with a straight vector on the flattened surface. Along this vector, a series of partially overlapping ROIs (12 mm radius, centers spaced 3 mm) was defined. From each ROI, neural RDMs were extracted, averaged, and entered into correlation-based RSA as described above. Resulting correlation coefficients were averaged across participants and plotted as a function of position on the dorsal–ventral axis.

## Data availability

A reporting summary for this article is available as a Supplementary Information file. The source data underlying Figs. 2d and 3b–d and Supplementary Figs 2b and 3 are provided as a Source Data file. The Source data file as well as neuroimaging data is deposited at the Open Science Framework ("6v8u4"). Stimulus materials and code are available upon reasonable request.

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

## Acknowledgements
This work was supported by the Provincia Autonoma di Trento and the Fondazione CARITRO (SMC). We thank Valentina Brentari for assistance in preparing the verbal stimulus material and with data acquisition.

## Author contributions
M.F.W. and A.C. developed the study design, M.F.W. collected and analyzed data, and M.F.W. and A.C. wrote the manuscript.

## Additional information

**Competing interests:** The authors declare no competing interests.

