## [Peer Review File · Nature Communications]

Reviewers' comments:

Reviewer #1 (Remarks to the Author):

The authors report an investigation aimed at a better understanding of the brain regions that support the representation of actions at a conceptual level. Specifically, they sought to identify regions in which patterns of BOLD activity discriminate amongst different actions but in which the same actions produce similar patterns whether they are described verbally or shown in a video. Participants performed tasks that involved either viewing simple meaningful actions or reading sentences about such actions. Crossmodal action classification was evidenced primarily in left lateral posterior temporal cortex. Further representational similarity analyses confirmed that activity in this region was related to offline measures of the semantic/conceptual (dis)similarities of the actions to each other. In contrast, unimodal action representations were widespread and overlapped in the fronto-parietal cortices, but in those regions did not display cross-modal correspondence.

The manuscript is clear and well-written. The study is highly rigorously designed and controlled, and the analyses are thorough, well-justified, and clearly described. The results provide compelling evidence in favour of the increasingly prevalent view that the lateral posterior temporal cortex plays a high-level role in action understanding – and sheds further light on how it does so. At the same time, the study casts further doubt on the widely-held proposal that the understanding of actions depends critically on the activity of motor-related regions in the frontal and parietal cortices. Overall, the study is a genuine technical and conceptual tour-de-force, and I find that I do not have even minor critiques or suggestions for improvements. I expect the paper to have a significant impact in the study of action understanding.

Reviewer #2 (Remarks to the Author):

This interesting study uses multi-voxel pattern analysis techniques to assess whether specific brain regions encode tool-related actions independent of the input format (videos or sentences), as well as assessing object- and person-directedness and semantic similarity.

There are a number of detailed analyses and controls that, for the most part, appear appropriately performed, and the study could make an important contribution pending several additional analyses and responses to several concerns.

The first concern is raised in part by the large overlap between the left pTC region reported here and left pTC regions previously reported to be activated by manipulable objects and body parts, including several studies from the authors' lab (e.g., Bracci et al, 2012; 2015). The current study is framed around actions/verbs, but of course the videos and sentences contain object-directed or non-object directed patients (hereafter, recipients). (In fact, the non-object-directed patients should be further divided into self-directed (self touching) versus other directed (symbolic gesturing) actions. As shown in Figure 3 c and d, materials referencing object directed versus person directed actions pattern differently in the pTC, so the question is raised of whether the apparent similarity of open, close, give and take; stroke and scratch; and agree and disagree is in fact at least in part due to the similarity of their recipients. Actions such as "he scratches the box" or "she gives her arm" would control for the inherent confound but cause other problems because of novelty and reduced meaning. The best solution may be to run a separate localizer session with the objects/body parts/human recipients used in the experiment to show that the regions identified in pTC as being concerned with actions does not substantially overlap the region that is activated for recipients. The investigators do show that an RSA with object semantics information from WordNet does not well fit the activation data, but the WordNet model is a language-based, taxomically-focused analysis tool that may not be well-suited to the relevant underlying

dimensions of object similarity to which this region is attuned —namely, event-based (thematic) relations (e.g., Kalenine, Peyrin et al., 2009). An RSA analysis that includes ratings for thematic role similarities (e.g., in terms of the agents and patients commonly associated with the actions) may reveal that pTC is indeed attuned to aspects of object semantics.

Another related concern is a theoretical one. The claim in the introduction is that action concepts are organized via propositional structures. In (potential) contrast, the claim in the discussion is that LPTC encodes “general and abstract” aspects of action. So the reader is left to infer that the authors believe that general and abstract aspects of action are actually propositional in nature. This claim is at the heart of a long-standing debate in the semantic literature, but ‘abstract’, non-embodied aspects of action need not be propositional; there are at least 2 other possibilities. One is an abstract trajectory representation (e.g., Wong, Goldsmith, & Krakauer, 2016) and the other is (again) an event-based representation that has been described as arising partly in pTC (see Mirman, Britt, and Landrigan, 2017, for review). The fact that the (propositional) WordNet based model adequately fit the data does not, of course, rule out other possible organizational structures.

The order effects analysis and analyses considering the effects of verbalization and visual imagery were commendable. There is a lingering concern that the sample size was too small to detect significant correlations between the verbalization measure and decoding accuracy, particularly as trend or near-trend level effects were observed in the 3 ROIs. There was also evidence for an order effect noted in the Supplement p. 1. It is important to control for these sources of variance and show that action dimensions of interest still result in robust decoding above and beyond any effects of verbalization and order.

The motivation for and description of the Bayesian whole brain analysis was unclear. This analysis showed evidence for cross-modal decoding not only in the left hemisphere ROI, but in homologous right hemisphere regions and in ITG. As shown in Figure 2a, a region in right pTC seems to approach significance for crossmodal classification based on a searchlight analysis. The mismatch between the two approaches is not well addressed.

Figure 1 C and 1 D are referenced in the text but were not provided. It seems that Figure 2A may have formerly been 1C in an earlier draft.

Axial sections should additionally be provided for Figure 2.

Responses to the Reviewers' comments:

Reviewer #1 (Remarks to the Author):

The authors report an investigation aimed at a better understanding of the brain regions that support the representation of actions at a conceptual level. Specifically, they sought to identify regions in which patterns of BOLD activity discriminate amongst different actions but in which the same actions produce similar patterns whether they are described verbally or shown in a video. Participants performed tasks that involved either viewing simple meaningful actions or reading sentences about such actions. Crossmodal action classification was evidenced primarily in left lateral posterior temporal cortex. Further representational similarity analyses confirmed that activity in this region was related to offline measures of the semantic/conceptual (dis)similarities of the actions to each other. In contrast, unimodal action representations were widespread and overlapped in the fronto-parietal cortices, but in those regions did not display cross-modal correspondence.

The manuscript is clear and well-written. The study is highly rigorously designed and controlled, and the analyses are thorough, well-justified, and clearly described. The results provide compelling evidence in favour of the increasingly prevalent view that the lateral posterior temporal cortex plays a high-level role in action understanding – and sheds further light on how it does so. At the same time, the study casts further doubt on the widely-held proposal that the understanding of actions depends critically on the activity of motor-related regions in the frontal and parietal cortices. Overall, the study is a genuine technical and conceptual tour-de-force, and I find that I do not have even minor critiques or suggestions for improvements. I expect the paper to have a significant impact in the study of action understanding.

We thank the Reviewer for the positive comments.

Reviewer #2 (Remarks to the Author):

This interesting study uses multi-voxel pattern analysis techniques to assess whether specific brain regions encode tool-related actions independent of the input format (videos or sentences), as well as assessing object- and person-directedness and semantic similarity.

There are a number of detailed analyses and controls that, for the most part, appear appropriately performed, and the study could make an important contribution pending several additional analyses and responses to several concerns.

The first concern is raised in part by the large overlap between the left pTC region reported here and left pTC regions previously reported to be activated by manipulable objects and body parts, including several studies from the authors' lab (e.g., Bracci et al, 2012; 2015). The current study is framed around actions/verbs, but of course the videos and sentences contain object-directed or non-object directed patients (hereafter, recipients). (In fact, the non-object-directed patients should be further divided into self-directed (self touching) versus other directed (symbolic gesturing) actions. As shown in Figure 3 c and d, materials referencing object directed versus person directed actions pattern differently in the pTC, so the question is raised of

whether the apparent similarity of open, close, give and take; stroke and scratch; and agree and disagree is in fact at least in part due to the similarity of their recipients. Actions such as “he scratches the box” or “she gives her arm” would control for the inherent confound but cause other problems because of novelty and reduced meaning. The best solution may be to run a separate localizer session with the objects/body parts/human recipients used in the experiment to show that the regions identified in pTC as being concerned with actions does not substantially overlap the region that is activated for recipients. The investigators do show that an RSA with object semantics information from WordNet does not well fit the activation data, but the WordNet model is a language-based, taxomically-focused analysis tool that may not be well-suited to the relevant underlying dimensions of object similarity to which this region is attuned —namely, event-based (thematic) relations (e.g., Kalenine, Peyrin et al., 2009). An RSA analysis that includes ratings for thematic role similarities (e.g., in terms of the agents and patients commonly associated with the actions) may reveal that pTC is indeed attuned to aspects of object semantics.

RESPONSE:

The Reviewer raises an important issue: the actions tested in our study were directed toward different types of recipients. Could it be that the representations identified in posterior temporal cortex encoded information about the recipients alone rather than information specific for actions? The Reviewer suggests two analyses to test this possibility:

The first is to test whether the region identified in posterior temporal cortex as being concerned with actions substantially overlaps with the regions that are activated for recipients (manipulable objects, persons, hands/arms). Unfortunately we do not have the possibility to run an additional fMRI localizer experiment. We therefore believe that the best solution is to compare the extent of the LPTC cluster found in our study with the cluster locations of comparable localizers (for tools, whole bodies, and hands) from the studies mentioned by the reviewer (Bracci et al., *J Neurophysiol*, 2012; Bracci et al., *JNS*, 2015; we also included Bracci & Peelen, *JNS*, 2013). The clusters identified in these studies show very similar peak locations across studies, which indicates strong anatomical reliability and suggests that the identified effects are suitable for comparison with our study.

Supplementary Figure 4 shows the outlines of the LPTC cluster identified in our study and the peak locations for contrasts that target representations of whole bodies (e.g. whole bodies vs. chairs), hands (e.g. hands vs. chairs), and manipulable objects (e.g. tools vs. chairs). The peaks of these contrasts lie more posterior and ventral relative to the boundary of the LPTC cluster. Two exceptions in the posterior/ventral part of LPTC are peaks from the contrasts hands/tools vs. animals (instead of hands/tools vs. chairs; which might explain why these locations differ from the other localizer locations). We conclude that the LPTC cluster found in our study is more anterior and dorsal relative to LOTC areas associated with the visual processing of persons, hands, and manipulable objects.

One might object that the standard localizers are not optimally suited for a comparison with our study because the localizers are based on univariate (rather than multivariate) methods and because they specifically target visual representations since they use object pictures as stimuli. For an additional comparison, we therefore also

considered two studies that identify areas in posterior temporal cortex using crossmodal searchlight MVPA of object categories (Fairhall & Caramazza, 2013: tools, fruit, clothes, mammals, and birds; Simanova et al., 2012: tools and animals). These studies identified clusters that were located at the ventral/anterior boundaries of LPTC (in inferior temporal gyrus). Since the peak locations were distant from the peak locations found in our study, we conclude that these findings do not support the assumption that our study identified information specifically about objects rather than actions.

The Reviewer also suggests to test a model of event-based (thematic) relations, e.g., using ratings for thematic role similarities (e.g., in terms of the agents and patients commonly associated with the actions). This could test whether the representational organization in LPTC is shaped by thematic rather than taxonomic object relations. This is generally an excellent suggestion and we are sympathetic with the view that LPTC contains representations that integrate objects based on event-based principles (e.g. *bottle* and *glass* are thematically linked by the action *pouring*). In our study, we do not investigate thematic relations between two varying objects but rather between a single actor and different recipients (actors do not vary with the actions; note also that we collapse across actors). It would be unhelpful to compute the thematic relations between the objects used as recipients (e.g. between *pencil case* and *elbow* or between *bottle* and *colleague*; note in this context that we collapse across objects used in different action scenarios such as pencil case and bottle, which renders a reasonable analysis of thematic relations between broad classes of recipients difficult) and we think that this is not what the Reviewer suggests. Rather, the idea seems to be to test whether the actions are similar in terms of the objects that are typically associated with the actions. However, we think that this is what our models of person- and object-directedness already capture. It would in principle be possible to use a more general model that describes whether the actions tested in our study *generally* target similar or dissimilar objects. However, it is not clear to us what the benefit would be to test this model rather than the models of person- and object-directedness. Our study was specifically designed to test person- and object-directedness as major determinants of action organization and it does not seem that insight would be gained by collapsing these dimensions to a single factor. Note also that we do not claim that person- and object-directedness are the only relevant types of recipients and concur with the Reviewer that other types of recipients should be investigated (e.g. self-directed actions, as suggested by the Reviewer).

In the revised version of our manuscript we included the comparison of LPTC with peak locations in the result section (p. 6, Supplementary Figure 3), we discuss the idea of thematic relations (see also our response to the second comment below), and we make clear that person- and object-directedness in fact capture two specific types of thematic roles – persons and inanimate objects as two major classes of action recipients.

Another related concern is a theoretical one. The claim in the introduction is that action concepts are organized via propositional structures. In (potential) contrast, the claim in the discussion is that LPTC encodes “general and abstract” aspects of action. So the reader is left to infer that the authors believe that general and abstract aspects

of action are actually propositional in nature. This claim is at the heart of a long-standing debate in the semantic literature, but ‘abstract’, non-embodied aspects of action need not be propositional; there are at least 2 other possibilities. One is an abstract trajectory representation (e.g., Wong, Goldsmith, & Krakauer, 2016) and the other is (again) an event-based representation that has been described as arising partly in pTC (see Mirman, Britt, and Landrigan, 2017, for review). The fact that the (propositional) WordNet based model adequately fit the data does not, of course, rule out other possible organizational structures.

RESPONSE:

We thank the Reviewer for highlighting this ambiguity and the seeming inconsistency between statements in the introduction and the discussion. Our results suggest that LPTC is sensitive to different basic aspects of actions that can be accessed via different stimulus types. However, the precise nature of action concepts (e.g. whether these aspects are best understood as components of propositional action structures or are encoded in an iconic or picture-like format) cannot be conclusively resolved by our study.

The Reviewer suggests two alternatives of how action information could be represented in an abstract manner. Following the idea of abstract trajectory representation, it is possible that aspects of actions are represented as basic movement trajectories. Such representations, however, have been associated with premotor and parietal rather than posterior temporal regions (Wong et al., 2016). While we cannot rule out that such representations could be possible in posterior temporal cortex, we think that due to the variability of action exemplars within each action condition our study is not appropriate for testing this idea in a convincing manner. Another possibility is that the similarity of action information in LPTC is due to event-based representations. As elaborated above, our study is not well suited to investigate thematic relations between objects. But what about thematic/event-based relations between actions? Thematic relations are thought to “reflect co-occurrence in the scenarios and event such as birthday parties or baking” (Mirman et al., 2017), and this principle does not only apply to objects (e.g. *mouse-cheese*) but also to actions (e.g. *aim-shoot*). It appears likely that event-based relations constitute an additional factor explaining representational similarity (in LPTC or other regions, such as temporoparietal cortex, as proposed by Mirman et al., 2017). But again, because of the high variance across action exemplars and the use of different contextual settings in the videos (breakfast and office), we think that our study is not well suited to test this idea directly. An exciting follow-up study could test whether actions belonging to the same overarching activity (e.g. *opening jar*, *spreading jam*) are represented in a similar manner, relative to taxonomically similar/dissimilar actions (e.g. *opening paint box*, *painting with brush*). Note that for actions (but not for objects), WordNet similarity indeed seems to capture aspects that resemble thematic relations (entailment relations, as e.g. *snore-sleep*).

In our revision, we explain in greater detail that our study was particularly designed to test the factors person- and object-directedness (and semantic similarity as conceptualized by WordNet as an exploratory factor) and that this does not rule out the existence of additional organizational principles in LPTC or other brain regions.

The order effects analysis and analyses considering the effects of verbalization and visual imagery were commendable. There is a lingering concern that the sample size was too small to detect significant correlations between the verbalization measure and decoding accuracy, particularly as trend or near-trend level effects were observed in the 3 ROIs. There was also evidence for an order effect noted in the Supplement p. 1. It is important to control for these sources of variance and show that action dimensions of interest still result in robust decoding above and beyond any effects of verbalization and order.

RESPONSE:

The Reviewer correctly points out that also the dimensions of interest (person- and object-directedness, action semantics) should be controlled for effects of verbalization, visual imagery, and session order. Unfortunately, it is not possible to generate models of these measures, which could be entered as additional predictors in the multiple regression RSA (because ratings were not collected for each action separately). However, it is possible to test whether the betas of the dimensions of interest show order effects or correlate with the ratings for verbalization and imagery. The underlying assumption is that the representational organization in LPTC could be influenced by order effects, verbalization, or visual imagery, e.g., imagining the specific features of object-directed actions (such as reaching and grasping movements) could bias the organization of actions along object-directedness.

To test whether the action dimensions of interest (person- and object-directedness, action semantics) could be affected by verbalization, visual imagery, and session order, we entered the beta values resulting from the multiple regression RSA into the control analyses as reported for crossmodal decoding accuracies (two-tailed independent t-tests, one-tailed Pearson correlations)). There were no significant effects of session order or significant correlations (see Supplementary Figure S5 for plots of the correlations):

Person-directedness:

session order (independent t test between groups): $t=-0.290$, $p=0.775$

verbalization: $r=-0.065$, $p=0.610$

imagery*: $r=-0.570$, $p=0.997$

verbal-sent corresp: $r=-0.160$, $p=0.755$

imagery-vid corresp: $r=-0.393$, $p=0.961$

Object-directedness:

session order (independent t test between groups): $t=-0.768$, $p=0.452$

verbalization: $r=0.143$, $p=0.268$

imagery: $r=-0.198$, $p=0.805$

verbal-sent corresp: $r=-0.023$, $p=0.539$

imagery-vid corresp: $r=-0.175$, $p=0.776$

Action semantics:

session order (independent t test between groups): $t=-1.664$, $p=0.113$

verbalization: $r=0.277$, $p=0.112$

imagery: $r=-0.061$, $p=0.604$

verbal-sent corresp: $r=0.156$, $p=0.250$

imagery-vid corresp: $r=-0.242$, $p=0.855$

* there appears to be a negative correlation between imagery and person-directedness. However, if we use a two-tailed correlation, the effect is not significant after FDR correction. The negative correlation is therefore likely to arise by chance.

The Reviewer mentions that we report a significant order effect in the Supplement p. 1. Just to clarify, this effect refers to the behavioral data: it shows that session order influenced the ratings for correspondence between videos and visually imagined actions (higher for “video first” group) and between sentences and verbalizations (higher for “sentence first” group). This significant interaction may suggest that the obtained rating scores are generally suitable for testing effects of verbalization and visual imagery. We clarified this in the supplemental material.

In addition, we made explicit in the revised manuscript that there were marginally significant trends for effects of verbalization. However, the whole-brain analysis did not reveal particularly increased trends for verbalization in LPTC in comparison to other brain regions; rather, several other regions outside LPTC show similar or stronger trends for verbalization. We therefore do not consider this marginal trend to be meaningful. Note that the correlation maps cannot be corrected for multiple comparisons using the procedures that were applied to the other reported maps (within and between group t tests).

The motivation for and description of the Bayesian whole brain analysis was unclear. This analysis showed evidence for cross-modal decoding not only in the left hemisphere ROI, but in homologous right hemisphere regions and in ITG. As shown in Figure 2a, a region in right pTC seems to approach significance for crossmodal classification based on a searchlight analysis. The mismatch between the two approaches is not well addressed.

RESPONSE:

We have clarified the motivation for and the description of the Bayesian whole brain analysis in the revised manuscript (see Figure legend to Supplementary Figure 1). Importantly, this analysis does not provide statistical measures of significance but likelihoods for or against the tested hypothesis in the data. The general purpose of the Bayesian model comparison was to provide an estimate of whether the absence of crossmodal decoding is meaningful or could be due to a lack of power in the data. The purpose of the whole brain analysis was to also provide likelihood estimates for other frontoparietal areas that were not covered in the ROI analysis. In other words, the mean accuracy map in Figure 2A shows several areas at or slightly above chance, but the map cannot tell whether the absent or weak effect is due to a true null effect or due to a lack of power. The Bayesian maps therefore add another layer of description to the mean accuracy map (and the statistical map on which the correction outlines are based on) by providing additional information about the likelihood for and against the absence of an effect.

It is important to point out that the whole brain maps cannot be corrected for multiple comparisons, and fluctuations around chance decoding would be expressed as Bayes factors **over or above** 1 [original text: 'as Bayes factors over or above 1' was a typo]. Because there is no possibility to correct the maps, the higher likelihoods for H1 in right LPTC and left ITG (with only small clusters showing BFs > 30; no BFs > 100; for comparison: BFs in left LPTC are between 100 and 100.000) cannot be interpreted as conclusive positive evidence for crossmodal effects in these

above
or below

areas. There is hence no mismatch between the Bayesian whole brain analysis and the mean accuracy map shown in Figure 2A.

Figure 1 C and 1 D are referenced in the text but were not provided. It seems that Figure 2A may have formerly been 1C in an earlier draft.

RESPONSE: Corrected.

Axial sections should additionally be provided for Figure 2.

RESPONSE: Done.

We thank the Reviewer for the constructive criticisms.

** See Nature Research's author and referees' website at www.nature.com/authors for information about policies, services and author benefits

This email has been sent through the Springer Nature Tracking System NY-610A-NPG&MTS

Confidentiality Statement:

This e-mail is confidential and subject to copyright. Any unauthorised use or disclosure of its contents is prohibited. If you have received this email in error please notify our Manuscript Tracking System Helpdesk team at <http://platformsupport.nature.com>.

Details of the confidentiality and pre-publicity policy may be found here

<http://www.nature.com/authors/policies/confidentiality.html>

Privacy Policy | Update Profile

DISCLAIMER: This e-mail is confidential and should not be used by anyone who is not the original intended recipient. If you have received this e-mail in error please inform the sender and delete it from your mailbox or any other storage mechanism. Springer Nature Limited does not accept liability for any statements made which are clearly the sender's own and not expressly made on behalf of Springer Nature Ltd or one of their agents.

Please note that Springer Nature Limited and their agents and affiliates do not accept any responsibility for viruses or malware that may be contained in this e-mail or its attachments and it is your responsibility to scan the e-mail and attachments (if any).

REVIEWERS' COMMENTS:

Reviewer #2 (Remarks to the Author):

The authors have adequately addressed all of the prior concerns. This should make an important contribution to the literature.